# Gender, Adverse Changes in Social Engagement and Risk of Unhealthy Eating: A Prospective Cohort Study of the Canadian Longitudinal Study on Aging (2011–2021)

**DOI:** 10.3390/nu17061005

**Published:** 2025-03-13

**Authors:** Sanaz Mehranfar, Gilciane Ceolin, Rana Madani Civi, Heather Keller, Rachel A. Murphy, Tamara R. Cohen, Annalijn I. Conklin

**Affiliations:** 1Food, Nutrition and Health, Faculty of Land and Food Systems, The University of British Columbia, Vancouver, BC V6T 1Z4, Canada; sanazmr@student.ubc.ca (S.M.); tamara.cohen@ubc.ca (T.R.C.); 2Collaboration for Outcomes Research and Evaluation, Faculty of Pharmaceutical Sciences, The University of British Columbia, Vancouver, BC V6T 1Z3, Canada; gilciane.ceolin@ubc.ca (G.C.); ranam77@student.ubc.ca (R.M.C.); 3Department of Kinesiology and Health Sciences, University of Waterloo, Waterloo, ON N2L 3G1, Canada; heather.keller@uwaterloo.ca; 4Schlegel-UW Research Institute for Aging, Waterloo, ON N2J 0E2, Canada; 5School of Population and Public Health, The University of British Columbia, Vancouver, BC V6T 1Z3, Canada; rachel.murphy@ubc.ca; 6Cancer Control Research, BC Cancer Research Centre, Vancouver, BC V5Z 1L3, Canada; 7Healthy Starts, BC Children’s Hospital Research Institute, BC Children’s Hospital, Vancouver, BC V5Z 4H4, Canada; 8Centre for Advancing Health Outcomes, Providence Health Care Research Institute, St. Paul’s Hospital, Vancouver, BC V6Z 1Y6, Canada; 9Edwin S.H. Leong Centre for Healthy Aging, Faculty of Medicine, The University of British Columbia, Vancouver, BC V6T 1Z3, Canada

**Keywords:** social relationships, fruits and vegetables, gender, longitudinal study, CLSA

## Abstract

Background: Social isolation is linked to survival and health. However, dietary effects of social activities, and gender differences, over time are unknown. Methods: A prospective study of adults (45+y) reporting daily fruit or vegetable (F/V) intake (at wave 1) from the Canadian Longitudinal Study on Aging (CLSA). Multivariable mixed logistic regression assessed changes in social isolation or breadth of social participation (wave 1 to 2) in relation to adverse changes in F/V (non-daily intake) at wave 3 in women and men. Results: Women who remained socially isolated between waves 1 and 2 had 85% higher odds of non-daily vegetable intake (OR 1.85 [95% CI: 1.32, 2.59]) and over twofold higher odds of non-daily fruit intake (2.23 [1.58, 3.14]), compared to reference (not isolated at waves 1 and 2). Higher odds of non-daily F/V intake were also observed for women who changed from isolated at wave 1 to not isolated at wave 2. Women and men who had less diverse social participation at waves 1 and 2 had 28–64% higher odds of non-daily F/V intake, compared to their counterparts with diverse social participation at both waves. Higher odds of non-daily fruit were also seen for women who had diverse social participation at wave 1 but reduced their diversity at wave 2 (1.35 [1.12, 1.62]). Conclusions: Results showed persistent social isolation impacted changes in F/V among women only, while limited breadth of social participation affected F/V intake in both genders. Further longitudinal research on the complexities of social engagement and eating behavior is warranted.

## 1. Introduction

By 2030, one in six people worldwide will be aged 60 or older [1]. The demographic shift in Canada will surge in 2031 when all baby boomers will have turned 65, with older adults comprising about 25% of the population by 2036 [2]. Aging in late adulthood frequently accompanies significant changes in the social environment, which includes a high degree of social isolation [3]. Almost one-third of Canadian older adults are at risk of social isolation [4]. Thus, remaining socially connected and socially active is viewed as the foremost emerging issue for older adults according to the International Federation on Ageing [5].

The extent to which a person engages in social activities where they interact with others in the community can affect health and well-being, particularly in late adulthood [6,7]. It is known that social integration is linked to higher odds of survival [8] and overall healthy aging [9]. Specifically, higher social participation is linked to better self-perceived health and reduced feelings of loneliness and dissatisfaction [10]. Conversely, social isolation is associated with a higher risk of mortality [11,12]; however, the evidence is limited by reliance on social isolation scores (e.g., the Shankar index) [13] that combine lone-living, infrequent social contact, no social participation, and not married [13,14,15,16,17,18]. Although social isolation and loneliness are often studied together or used interchangeably, social isolation is an objective structural indicator (i.e., absence of others) whereas feelings of loneliness are a subjective functional indicator (i.e., an emotional state) [11,19]. Scope exists to disentangle social isolation in healthy aging research given that structural social ties show stronger effects on health outcomes [11] and risk factors [20,21,22,23].

Social relationships are theorized to impact health through multiple pathways [24], including health-related behaviors such as diet [25]. Indeed, the social environment is known to influence food intake and diet [26]. Specifically, social isolation has been linked to lower dietary quality [7,27], reduced intake of fruits and vegetables (FV) [14,28,29], lower dietary micronutrient intake [30], and lower dietary variety [31]. By contrast, social participation is associated with lower nutritional risk [32], better diet quality [33], and higher FV intake [34]. Much of the current literature, however, is limited to cross-sectional studies that lack temporality between the social factors and diet and/or lack assessment of alterations in social integration over time, such as becoming socially isolated or reducing social participation. To date, no study has examined the changes in social participation in relation to changes in healthful eating indicators among aging adults [35]. Moreover, nutritional epidemiology research rarely disaggregates data on women and men despite the fact that gender dynamics significantly influence both eating habits [36] and social participation [10,37]. Thus, great scope exists to investigate whether changes in social isolation and in the breadth of social participation reduce or increase dietary fruit or vegetable intake and, especially, to understand whether women and men differ in vulnerability to the strength of impact.

This study leverages three waves of data from a Canadian aging cohort to prospectively examine whether adverse changes in social isolation (lack of activities) or breadth of social participation (variety of activities) are associated with the development of unhealthful eating habits among older women and men. We tested the hypothesis that (1) adverse changes (e.g., becoming socially isolated or reducing the breadth of social participation) will be associated with adverse change in healthful eating (i.e., a decline in daily fruit or vegetable (F/V) intake) and that (2) associations will differ by gender. Our approach addresses critical knowledge gaps in the aging literature by considering both the heterogeneous dynamics of social ties over time and the temporal separation of social tie transitions and dietary behavior changes.

## 2. Materials and Methods

### 2.1. Study Population and Eligible Sample

We used three waves of data from 30,097 predominantly White, non-institutionalized, middle-age and older adults (45–85 years) in the Canadian Longitudinal Study on Aging (CLSA) Comprehensive Cohort—a stratified random sample of community-dwelling individuals [38]. This cohort had self-reported data and anthropometric measurements collected from English- or French-speaking individuals residing within a 25–50 km of 11 specified data collection sites during three waves. This cohort excluded residents of the three territories, federal First Nations reserves and other First Nations settlements in the provinces, remote regions, full-time members of the Canadian Forces, and individuals living in institutions or with cognitive impairment at the time of recruitment [39,40]. To address the under-representation of individuals with lower education levels and socioeconomic status in population-based studies, special efforts were undertaken to oversample specific areas identified through census data [41]. The detailed CLSA study design is available elsewhere [39,40,41]. In order to assess decline in healthful eating (non-daily F/V), our eligible sample included CLSA participants who reported consuming fruits (n= 21,178, 74%) or vegetables (n = 18,333, 64%) at least one time per day at baseline (wave 1: 2011–2015) and, also, who had information on social participation and other covariables (respectively, n = 18,413 or n = 15,999) (details of sample size in Appendix A). We followed the Strengthening the Reporting of Observational Studies in Epidemiology (STROBE) guidelines for reporting observational studies [42].

### 2.2. Changes in Fruit or Vegetable Intake

Our primary outcome was the adverse change from healthful eating (daily F/V intake) at wave 1 (2011–2015) to less healthful eating (non-daily F/V intake) at wave 3 (2018–2021); that is, the development (new-onset) of less healthful eating over 6-year follow-up. Thus, we considered less healthful eating as fruit intake less than once per day (non-daily fruit) and vegetable intake less than once per day (non-daily vegetable) using dietary data collected by CLSA’s Short Diet Questionnaire (SDQ). The SDQ is a validated 36-item non-quantitative food frequency questionnaire (FFQ) [43] that was developed and used in the CLSA to assess usual consumption frequencies (last 12 months) of key nutrients and foods of importance for health promotion and chronic disease prevention in community-dwelling middle-age and older adults [43]. Five relevant questions (fruit, green vegetables, potatoes, carrots, and other vegetables) were available for analysis. We defined healthful eating (at least daily intake of fruits or vegetables) [14] based on frequency rather than serving size in accordance with Canada’s *Food Guide* [44] that does away with portion sizes. CLSA responses for consumption of food items were daily, weekly, monthly, or yearly, followed by another question of “how many times” for that frequency response (e.g., twice a day, three times a week, once a month). CLSA converted all SDQ responses into times per day that we dichotomized for fruits (0 = less than daily, 1 = daily), green vegetables (0 = less than daily, 1 = daily), potatoes (0 = less than daily, 1 = daily), carrots (0 = less than daily, 1 = daily), and other vegetables (0 = less than daily, 1 = daily). The vegetable outcome variable was based on any of the four possible items being coded as daily intake (details of coding method in Appendix A). We excluded extreme values of fruit or vegetable intake that may result from measurement error or non-representative unique observations. To maintain consistency in our approach to outlier exclusion, we followed the National Cancer Institute Dietary Screener Questionnaire in the US NHANES 2009–2010 method [45] to exclude outliers in daily intake (>8 times/day for fruit; >5 times/day for vegetables including carrots, green vegetables, and others; and >3 times/day for potatoes) before constructing binary F/V variables at wave 1 (to restrict the sample) and wave 3 (to assess outcome).

### 2.3. Changes in Social Isolation and Breadth of Social Participation

We assessed changes in social isolation and in breadth of social participation based on CLSA data on the number of different social activities. CLSA asked participants a group of questions about the frequency of social activities in the last year across eight settings: *‘in the past 12 months, how often did you participate in [family, religious, sports, educational/cultural, social clubs, charity, neighborhood, and other recreational activities]?*’. For each setting, we considered regular participation to be at least once a month or more (coded as 1) versus less than monthly (coded as 0), similar to previous research [20,21,46], and then counted the number of activities at waves 1 and 2 to create a total activity score with good internal consistency (Cronbach’s α of 0.66). We considered the lower extreme of zero or 1 activity per month (scores combined due to few observations of zero) to assess social isolation at both waves; we, also, classified the upper extreme of 5 or more activities per month to assess breadth of social participation at both waves [20,21,46], which represented the highest tertile of the total activity score. The social isolation transition variable was classified as: (1) remained not isolated (no change in 2 or more activities at both waves, reference); (2) remained isolated (no change in 0 or 1 activity at both waves); (3) became socially isolated (transition to 0 or 1 activity); and (4) became not isolated (transition to 2 or more activities). The breadth of social participation transition variable was classified as: (1) remained diverse (no change in ≥5 activities at both waves, reference); (2) remained less diverse (no change in <5 activities at both waves); (3) became less diverse (transition from ≥5 activities to <5 activities); and (4) became diverse (transition from <5 activities to ≥5 activities). The social participation transition variable was weakly correlated with the social isolation transition variable (Cramer’s V = 0.20).

### 2.4. Covariables

Study duration was calculated in months between wave 1 and wave 3. Appendix A shows the Directed Acyclic Graph (DAG) we created with DAGitty software (version 3.0; Nijmegen, GE, The Netherlands) [47] to include theoretical, plausible, and known confounders [48,49,50,51,52,53,54,55,56]: (1) biological factors (age, body mass index (BMI) (proxy of energy intake), one or more chronic conditions, including depression); (2) socioeconomic status indicators (education (post-secondary graduation [university degree], some post-secondary education, secondary school graduation [high school diploma], below secondary school graduation [less than high school diploma]), household income (≥CAD 150,000, ≥CAD 100,000 to CAD 149,999, ≥CAD 50,000 to CAD 99,999, ≥CAD 20,000 to CAD 49,999, <CAD 20,000, do not know/refuse), wealth (home-owner, renter), and rural/urban location); (3) behavioral factors (at least 7 h/day of sleep); and (4) provincial factors (gross domestic product (GDP), public spending, food insecurity, and consumer price index for F/V). Provincial covariables were obtained from national statistics based on the average recruitment date of participants in each province.

### 2.5. Sex and Gender Considerations

At wave 1, participants answered ‘are you male or female?’, with responses representing biological sex and/or gender identity. Additional CLSA data on “gender identity” (male, female, transgender, transman, transwoman, genderqueer, and other) were collected at wave 2; both CLSA variables very strongly correlated (Cramer’s V = 0.998), with very few non-cisgender observations excluded (n = 40). Thus, we interpret results for women and men as reflecting cisgender identity, gender roles, gender relations, and institutionalized gender [57].

### 2.6. Statistical Analysis

Descriptive analysis used means (SD) and frequencies (%) to describe sample characteristics across the categories of each transition variable and applied CLSA survey inflation weights to produce estimates that represent the entire population of Canadians aged 45 to 85 residing in the vicinity of the data collection sites [41]. Main analysis examined the prospective associations of social participation or social isolation changes with developing non-daily fruit or non-daily vegetable intake in aging women and men. Multivariable mixed-effect logistic regressions (new-onset non-daily intake) used a random coefficient model (individuals nested within geographical areas) to account for the natural nesting of CLSA data and included an interaction term (transition variable x gender) to generate ORs specific to women and men. We ran duplicate regressions using reverse coding of the interaction term (women = 0 and men = 1; women = 1 and men = 0) to provide estimates for each reference group from the same model. Inferential statistics did not include analytic weights because CLSA does not provide sampling weights at both the primary sampling unit (Level 1) and the geographical level (Level 2), which are necessary for a fully-weighted multilevel model [58]. Model fit was tested using likelihood ratio tests and Akaike and Bayesian information criteria. Post-estimation of regression coefficients then calculated the average predicted probability of non-daily intake of fruits or vegetables associated with each category of transition (STATA ‘margins’ command), which were plotted for ease of interpretation (‘marginsplot’). Final results are reported as odds ratios (OR), or average predicted probabilities, with 95% confidence intervals (95% CI). Finally, sample sizes varied by outcome (vegetable n = 15,999; fruit n = 18,413). Analyses were carried out using Stata/SE 18.

Sensitivity analyses of main results were further conditioned on other potential confounders or independent predictors based on previous literature, including oral health [59], smoking [60], alcohol consumption [61], and, for women, reproductive variables (pregnancy, menopausal status, and hormone replacement therapy usage) [62]. For vegetable intake, we additionally excluded potatoes from the list of vegetable items. Final models were also adjusted for other wave 1 social relationships (e.g., living arrangement, marital status, and social network) to determine the independent effects of each transition variable. To assess the robustness of our findings regarding the impact of the COVID-19 lockdown on dietary behavior, we further excluded participants interviewed from 1 April 2020 onwards similar to another Canadian nutrition study [63].

## 3. Results

The average duration between CLSA waves 1 and 3 was 70.21 months (SD 5.81). For the smallest analytic sample (vegetable outcome), the mean age was 62 years (SD 9.9) and 59.0% were women. The majority of the sample remained not socially isolated between the first two waves of the CLSA cohort (91.64% of women and 89.53% of men). Nevertheless, 5.6% of women and 7.2% of men experienced adverse transitions, either remaining socially isolated or becoming socially isolated over time. Only about one-third of the sample’s social participation remained diverse (33.5% of women and 28.7% of men), and a much larger proportion experienced adverse changes (54.8% of women and 60.3% of men remained less diverse or became less diverse). Around 1 in 5 aging adults in our sample developed less healthful eating over the 6-year study duration, with 23% reporting intakes of vegetables less than once per day and 17.0% reporting intakes of fruits less than once per day.

Table 1 and Table 2 show an uneven distribution of sociodemographic characteristics across transitions in, respectively, social isolation and social participation over 3 years for women and men who reported daily intake of vegetables at wave 1. Variation in the development of non-daily vegetable intake across all the transitions in social isolation was more notable than across the transitions in breadth of social participation, with a more pronounced variation observed among men (Table 2) than among women (Table 1). Sample characteristics for the sample of fruit intake are given in Appendix A.

### 3.1. Social Isolation and Social Participation Transitions Associated with Reduced Vegetable Intake by Gender

Final models indicated that specific social isolation transitions were prospectively associated with subsequent non-daily vegetable intake among women only (Table 3). Specifically, women remaining socially isolated had 85% higher odds of new-onset non-daily vegetable intake (OR 1.85 [95% CI: 1.32, 2.59]), compared to women who remained not isolated. Women who became not socially isolated also had higher odds of new-onset non-daily vegetable intake (OR 1.38 [95% CI: 1.03, 1.84]), compared to the referent. No association was observed among men. By contrast, both women and men maintaining less diverse social participation (<5 activities/month at both waves) had higher odds of new-onset non-daily vegetable intake (respectively, 1.30 [95% CI: 1.15, 1.47]) and (1.29 [95% CI: 1.13, 1.46]), compared to counterparts remaining diverse in social participation (Table 4).

Sensitivity analysis of additional confounders did not alter the significant findings for women remaining socially isolated (Appendix A) or results for both women and men remaining less diverse in their social participation (Appendix A). Notably, results for women becoming not isolated were attenuated by adjustment for oral health and other social ties. To check robustness of findings to the influence of COVID-19 on eating habits, we excluded participants interviewed from 1 April 2020 onwards and found associations remained largely unchanged or even strengthened (in men for social isolation transitions) (Appendix A).

Figure 1 illustrates greater variation and more pronounced gender differences in the average predicted probability of reduced vegetable intake associated with distinct social isolation transitions (panel A) than social participation transitions (panel B) over time. The highest predicted probability of reduced vegetable intake occurred among women remaining socially isolated (28%), but among men becoming not socially isolated (37%). The lowest predicted probability of reduced vegetable intake for both women (18%) and men (31%) was observed among those remaining not socially isolated. The predicted probability of reduced vegetable intake for social isolation and social participation transitions was independent of other structured social ties (Appendix A).

### 3.2. Social Isolation and Social Participation Transitions Associated with Reduced Fruit Intake by Gender

Final models indicated that specific social isolation transitions were prospectively associated with subsequent non-daily fruit intake among women only (Table 5). Specifically, women remaining socially isolated had over twofold higher odds of new-onset non-daily fruit intake (OR 2.23 [95% CI: 1.58, 3.14]), compared to women who remained not isolated. Women who became not socially isolated also had higher odds of new-onset non-daily fruit intake (OR 1.77 [95% CI: 1.33, 2.37]), compared to the referent. No association was observed among men. By contrast, women and men maintaining less diverse social participation had higher odds of new-onset non-daily fruit intake (respectively, 1.64 [95% CI: 1.43, 1.88]) and (1.28 [95% CI: 1.12, 1.46]) and, also, women whose participation became less diverse (1.35 [95% CI: 1.12, 1.62]), compared to counterparts remaining diverse in social participation (Table 6).

Sensitivity analysis of additional confounders did not alter the main findings for women or men (Appendix A), with two exceptions. Adjusting for oral health and alcohol consumption strengthened results for men remaining or becoming socially isolated (Appendix A). The exclusion of participants interviewed during the COVID-19 lockdown did not affect the results for any of the exposures (Appendix A).

Figure 2 illustrates variations in the average predicted probability of reduced fruit intake associated with a change in social isolation that was more pronounced for women than for men (panel A). The highest predicted probability of reduced fruit intake occurred among women remaining socially isolated (25%) and the lowest probability was among women remaining not socially isolated (13%). Variation in the average predicted probability of reduced fruit intake associated with social participation transitions was more pronounced among women than men (panel B). The predicted probability of reduced fruit intake for social isolation and social participation transitions was independent of other structured social ties (Appendix A).

## 4. Discussion

This population-based prospective study revealed that specific transitions in both social isolation and breadth of social participation were significantly associated with developing less healthful eating among aging Canadians, and that associations were gendered. Although we did not find that becoming socially isolated reduced healthful eating, we did find that persistent social isolation and reduced breadth of social participation were both associated with a decline in healthful eating among women. We also found that maintaining less diverse social participation was also associated with a decline in healthful eating among women. However, our findings for men did not support our hypothesis that becoming socially isolated or reducing the breadth of social participation would be associated with a decline in the daily F/V intake. Nevertheless, we did show that maintaining less diverse social participation was associated with non-daily F/V intake among men. Our finding that becoming not socially isolated was associated with reduced healthful eating among aging women was unexpected and indicated that the transition out of social isolation may not be protective of diet quality for this subpopulation. Finally, an important finding from this research is that the dietary effect of changes in both social isolation and social participation was robust to other factors and, also, independent of other social relationships.

### 4.1. Relevance to Previous Research

In general, social relationships (both present and absent) are linked to longevity outcomes [8,11] and are also known to influence diet quality, particularly FV consumption, among older adults [31,64,65]. Recent research illustrates that aging women more than men, particularly of the oldest age and socioeconomic disadvantage, were more likely to become socially isolated or less diverse in social participation [46]. Yet, there is an evidence gap on whether alterations in an older person’s social environment, as a disruption in our homeostatic need for social connection, are true risk factors of healthful eating among aging adults. Our recent systematic review of longitudinal research on changes in social relationships and FV intake identified a large knowledge gap as current research only focuses on marital transitions and changes in FV intake [35]. This study is the first, to the best of our knowledge, to examine distinct alterations in social isolation and in the diversity of social participation as determinants of two indicators of healthful eating among middle-aged and older women and men in Canada.

Existing cross-sectional research indicates that social isolation is associated with poor health behavior profiles, such as low FV intake [29], which contributes to poor diet quality [25]. Across Europe, adults reporting high levels of social isolation were more likely to be physically inactive and to consume fewer fruits or vegetables on a daily basis compared to those reporting low/moderate levels of social isolation [14]. However, data were not disaggregated for women, transitions in isolation were not assessed, and social isolation was analyzed as a composite index, making results difficult to compare with this study. Indeed, the majority of health literature on social relationships continues to use composite scores for social isolation [66], which not only obscures intervention targets for tailored social prescribing but also masks potential gender differences for redressing health inequity. In a cross-sectional study of British older adults, lower levels of friend contact were associated with reduced variety of fruits and of vegetables in a graded trend for both genders (though more pronounced among men), and weekly family contact was also associated with women’s healthful eating indicators [31]. This study showed only women who remained socially isolated over three years were more likely to reduce their healthful eating in the subsequent three years (independent of marital status and living arrangement), which appears consistent with previous research.

The gender-specific dietary effect of remaining socially isolated among aging adults may be explained by differences in access to resources due to gender roles, relations, and institutional structures [67]. Women, who live longer than men, are more likely to live alone and to be more constrained by gender inequity that disproportionately shapes the more limited social (and economic) resources and opportunities available (and allowed) for women to support healthful behaviors and achieve good health [68]. Historically, women have had limited access to financial and social resources that constrain their ability to fully participate in society. For example, until the mid-1960s, Canadian women’s financial security was closely tied to their relationships with men, reinforcing economic dependency and unequal access to resources [69]. This dependency, coupled with women’s historically limited access to and control over family or household resources compared to men, not only reflects but also reinforces their unequal opportunities for full social and economic participation. Following divorce, women often experience more chronic financial strain, such as a decrease in household income and greater risk of poverty, that has negative consequences for women’s health [70], and especially for FV intake given FV are more costly food items per kcal. Other explanations for the observed decline in healthful eating among women who remained socially isolated include less social control over eating behaviors [71], absence of gendered social norms of women eating healthy [72], or lack of different types of social support for healthy behaviors [24].

A surprising and seemingly contradictory finding of this study was the adverse dietary consequences for women who transition out of social isolation. To understand this unexpected finding, we investigated the eight component activities for this transition category (Appendix A) and observed that women who became not socially isolated added monthly sport and other recreational activities (e.g., hobbies, gardening, poker, bridge, and cards). We, therefore, postulate that transitioning into regular recreational activities, which commonly involve sedentary games, may be accompanied by unhealthful snacking during social gatherings that replaced daily F/V intake. It is known that card-related games in a gambling setting are correlated with alcohol consumption, which might increase snack intake [73]. Similarly, while data on the type of sport are unavailable, golf is a popular activity among older adult White Canadians [74], which is often accompanied by alcohol consumption [75] that is known to pattern with poorer dietary habits [61]. Thus, this study demonstrates that transitioning out of social isolation may not offer a protective benefit for healthful eating among aging women and may depend on the setting of women’s social engagement. Findings need to be replicated in other cohorts and more nutrition research on this transition out of social isolation is needed.

An added novelty of this study is the assessment of the breadth of social participation. Other literature on social participation indicates that engaging in frequent social activities is linked to better psychosocial and health-related outcomes [76]; specifically, frequent involvement in diverse social activities can positively impact health, reduce loneliness, and improve life satisfaction [10]. Some nutrition studies indicate that higher levels of social participation are correlated with lower nutritional risk in older adults [32,77] and higher FV consumption, especially among lone-living older adults [34]. This study demonstrated that a persistent lack of breadth in social participation (maintaining less diverse social activities each month) was associated with reduced healthful eating, using both FV indicators among women and among men in Canada. Importantly for women, loss of breadth in social participation (becoming less diverse) was also associated with loss of daily fruit intake over time. In our post hoc exploration, we observed that women in this transition category disengaged from regular neighborhood, community, or professional association activities, volunteer work, charity work, educational or cultural activities, and sports (Appendix A). Broadly, our results support the notion that different types of relationships may fulfill distinct health-related needs, contributing to specific benefits [78]. In parallel to dietary guidelines advocating for FV variety to ensure nutrient balance, a corresponding recommendation of regular engagement with a variety of relationships and social roles may benefit different social needs and, in turn, diet [78].

Overall, this study revealed a clear gendered pattern of stronger associations between specific transitions in social isolation or breadth of social participation and healthful eating indicators among aging women. Women generally have larger networks [79,80] that extend beyond marriage and tend to rely on their friendship ties over time more than men [81]. Women’s social networks are more diverse and serve a wider range of functions compared to those of men [82]; thus, the broader social environment may be more impactful for women’s health-promoting behaviors. Our study showed that both social isolation and breadth of social participation transitions appear to matter for shaping women’s FV intake over time. It is suggested that participation in social relationships promotes health behaviors by providing opportunities for sociability, meaningful roles, and shared norms [24]. So, diverse social relationships reflect different gender roles, which can influence eating behaviors by assigning women caregiving responsibilities, such as food preparation and meal planning, or social responsibilities such as community organizing. As a result, women are more likely to engage in shared food practices and social eating routines. When women experience reduced social participation or become isolated, they lose access to their multiple gender roles that, then, disrupt their dietary habits. In contrast, men’s eating behaviors are less closely tied to caregiving roles and social expectations around food [83], making them less affected by changes in social connectedness, regardless of their cooking skills. Additionally, aging brings various challenges, such as financial stress or other health-related stressors, which disproportionately affect women who typically respond to stress through more pro-social strategies [84]. Hence, the lack of or reduced variety of social contacts could more negatively impact older women’s diets through compromised coping mechanisms. Our gender-specific findings on the relationship between social tie changes and changes in FV intake warrant further investigation to identify the influence of gender roles and societal norms that may shape how social transitions impact eating habits in older women.

Canada’s vision for healthy aging emphasizes social connectedness and healthy eating [85], and our findings offer actionable insights for designing gender-responsive interventions that promote both. Policies can prioritize increasing social engagement among older adults through community-based programs that encourage participation in diverse social activities. These programs should address barriers to engagement—such as mobility issues, transportation, and social stigma—while enhancing facilitators like accessible venues and inclusive programming. Incorporating nutritional support within these initiatives could further amplify their impact. For example, providing tailored nutritional information, free or subsidized meals, community gardens, and cooking workshops can directly support healthier eating behaviors. Indeed, fostering social participation is not only linked to improved dietary habits but also presents a valuable opportunity to deliver targeted nutritional interventions. Given that diet is a modifiable risk factor for chronic conditions, integrating social and nutritional support in community settings is essential for improving the well-being of aging populations. Furthermore, recognizing gender differences in how social activities influence eating behaviors is crucial for designing effective, equitable interventions. These efforts can help ensure that aging Canadians remain socially connected while benefiting from improved nutrition and overall health, ultimately supporting healthier aging trajectories.

### 4.2. Methodological Considerations

Participant recall errors in the frequency of fruit and vegetable intake were likely to result in non-differential misclassification bias in the outcome that would bring our estimates towards the null and underestimate associations. Self-reported frequency of social activity in the past year may also be prone to recall bias; however, such misclassification would be the same at all CLSA waves and, therefore, would not likely affect the exposure assessment of social isolation or social participation transitions over time. To reduce recall bias in dietary and social data, future studies could employ more objective collection methods, possibly using monitoring technologies; quantitative approaches to dietary assessment methods would also allow for calculating total energy intake. Results were limited by residual confounding due to imperfect measurement of covariables and unmeasured confounding due to missing dietary data on total energy intake, serving sizes, and food preparation skills. Given our dietary data were collected for one-third of the sample during the COVID-19 pandemic, the pandemic may have introduced reporting bias and altered our estimate of changes in healthy eating; the pandemic may also have altered some of the social activities following our exposure assessment in waves 1 and 2. Nevertheless, our sensitivity analysis that excluded participants interviewed from 1 April 2020 onwards (during the COVID-19 lockdown) demonstrated our results are robust. Lastly, as the CLSA cohort is predominantly White (96%), cisgender (99%), and heteronormative (98%) population, our findings cannot be generalized to more diverse populations or other settings.

Nonetheless, this study offers numerous strengths. These include two indicators of healthful eating, assessment of distinct transitions in both social isolation and breadth of social participation, adjustment for multiple known confounders, consideration of provincial sociopolitical differences, advanced statistical techniques to account for nested data, and application of a gender perspective. This study makes a novel contribution to the social relationship and health literature by examining alterations in the social environment and assessing a single measure of social isolation based on the absence of monthly social activity (independent of marital status, living arrangement, and social network size). Additionally, our analytic strategy of a target trial-emulated framework enabled a more rigorous estimation to support causal inference by ensuring temporality between changes in both exposures and outcomes among participants who were free of the outcome at wave 1 [86].

## 5. Conclusions

This prospective population-based study revealed that distinct changes in both social isolation and breadth of social participation were associated with indicators of healthful eating among aging adults in Canada, and that these associations were gendered. We found that persistent lack of regular social activity and both the persistent lack and loss of diversity in social activities had an impact on both indicators of healthful eating among women in particular. Only the persistent lack of diverse social activities appeared to shape healthful eating among men. The complexity of changes in the social environment on changes in diet among aging adults requires further investigation, especially through a gender lens.

## Figures and Tables

**Figure 1 nutrients-17-01005-f001:**
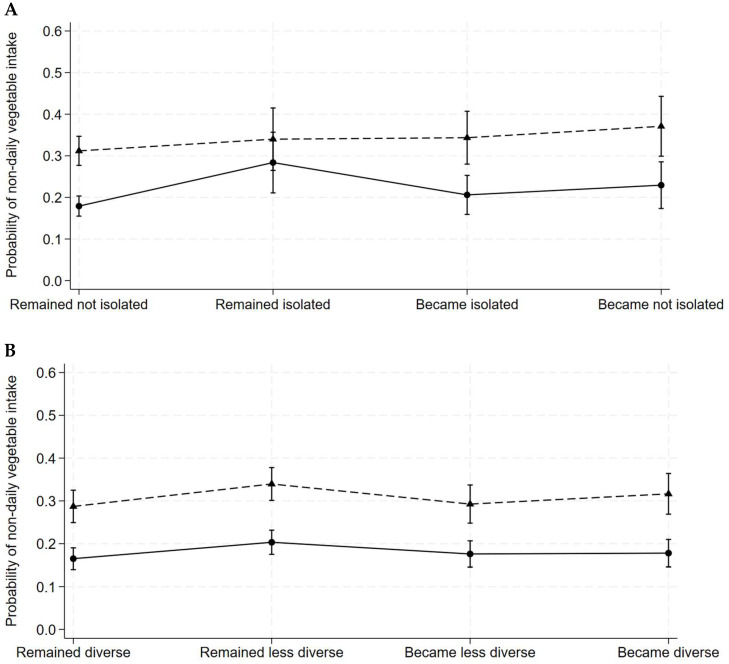
Average predicted probability of non-daily vegetable intake associated with social isolation transitions and social participation transitions among women and men in the Canadian Longitudinal Study on Aging (2011–2021). Dash line, men; solid line, women. (**A**), social isolation transitions; (**B**), social participation transitions. The social isolation transition variable is classified as: (1) remained isolated (no change in 0 or 1 activity/month at both waves); (2) remained not isolated (no change in 2 or more activities/month at both waves); (3) became isolated (transition to 0 or 1 activity/month); and (4) became not isolated (transition to 2 or more activities/month). Social participation transitions are classified as: (1) remained diverse (no change in ≥5 activities/month at both waves); (2) remained less diverse (no change in <5 activities/month at both waves); (3) became less diverse (transition from ≥5 activities to <5 activities/month); and (4) became diverse (transition from <5 activities to ≥5 activities/month).

**Figure 2 nutrients-17-01005-f002:**
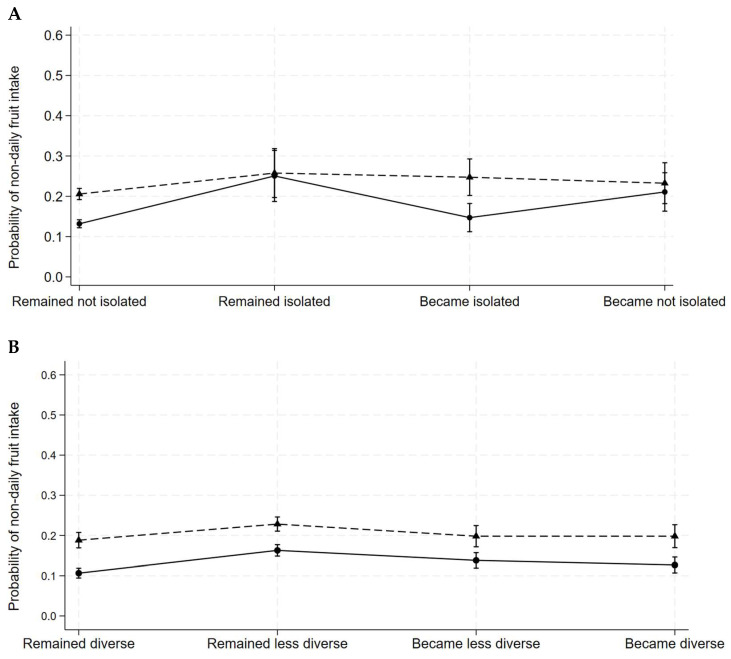
Average predicted probability of non-daily fruit intake associated with social isolation transitions and social participation transitions among women and men in the Canadian Longitudinal Study on Aging (2011–2021). Dash line, men; solid line, women. (**A**), social isolation transitions; (**B**), social participation transitions. The social isolation transition variable is classified as: (1) remained isolated (no change in 0 or 1 activity/month at both waves); (2) remained not isolated (no change in 2 or more activities/month at both waves); (3) became isolated (transition to 0 or 1 activity/month); and (4) became not isolated (transition to 2 or more activities/month). Social participation transitions are classified as: (1) remained diverse (no change in ≥5 activities/month at both waves); (2) remained less diverse (no change in <5 activities/month at both waves); (3) became less diverse (transition from ≥5 activities to <5 activities/month); and (4) became diverse (transition from <5 activities to ≥5 activities/month).

**Table 1 nutrients-17-01005-t001:** Sample characteristics across social isolation transitions among aging women and men in the eligible sample consuming daily vegetable intake in the Canadian Longitudinal Study on Aging (2011–2021).

Social Isolation Transitions	Mean (SD) Age (Years)	Highest Education ^1^	Highest Household Income ^2^	Home-Owner	Urban Location	BMI (kg/m^2^)	No Chronic Condition	Not Sleep Deprived	Non-Daily Vegetable Intake
				Women					
Remained not isolated (n = 8615)	59.09 (9.83)	6937 (67.5%)	1390 (17.1%)	7524 (87.4%)	7729 (92.2%)	27.52 (5.89)	463 (6.6%)	5503 (63.0%)	1532 (16.1%)
Remained isolated (n = 163)	63.56 (10.42)	98 (39.7%)	8 (9.9%)	110 (67.6%)	142 (93.2%)	28.05 (5.78)	8 (4.4%)	94 (51.1%)	53 (27.7%)
Became isolated (n = 366)	59.72 (11.18)	250 (51.8%)	34 (8.5%)	281 (77.2%)	324 (94.1%)	28.90 (6.84)	9 (1.5%)	197 (54.7%)	84 (26.1%)
Became not isolated (n = 257)	58.50 (8.7)	162 (47.7%)	25 (9.2%)	192 (70.1%)	233 (95.1%)	28.20 (6.72)	14 (14.5%)	149 (54.4%)	66 (19.1%)
				Men					
Remained not isolated (n = 5907)	58.82 (9.92)	4989 (70.9%)	1371 (23.4%)	5341 (88.0%)	5330 (94.4%)	27.92 (4.55)	429 (9.2%)	3877 (64.2%)	1754 (27.7%)
Remained isolated (n = 181)	61.77 (9.06)	130 (62.0%)	14 (5.9%)	145 (78.0%)	164 (93.0%)	28.89 (5.79)	5 (2.6%)	103 (59.8%)	66 (29.4%)
Became isolated (n = 292)	59.12 (8.79)	234 (68.2%)	41 (18.9%)	236 (75.9%)	258 (95.7%)	28.19 (5.65)	26 (9.1%)	181 (64.6%)	102 (34.6%)
Became not isolated (n = 218)	58.63 (10.62)	159 (49.8%)	30 (9.6%)	177 (75.1%)	191 (93.9%)	28.96 (5.30)	14 (3.6%)	136 (55.7%)	84 (43.9%)

Descriptive statistics, including percentages (%) and means (SD), were calculated using CLSA survey inflation weights. All variables were reported at baseline, except for vegetable intake at follow-up 2. ^1^ The highest education level was post-secondary graduation [university degree]. ^2^ The highest income was ≥CAD 150,000.

**Table 2 nutrients-17-01005-t002:** Sample characteristics across social participation transitions among aging women and men in the eligible sample consuming daily vegetable intake in the Canadian Longitudinal Study on Aging (2011–2021).

Social Participation Transitions	Age (Years)	Highest Education ^1^	Highest Household Income ^2^	Home-Owner	Urban Location	BMI (kg/m^2^)	No Chronic Condition	Not Sleep Deprived	Non-Daily Vegetable Intake
				Women					
Remained diverse (n = 3152)	60.68 (10.18)	2675 (74.7%)	521 (18.9%)	2817 (89.8%)	2856 (92.6%)	27.36 (5.75)	160 (6.6%)	2073 (66.9%)	503 (14.6%)
Remained less diverse (n = 3857)	58.43 (9.7)	2852 (61.1%)	574 (15.1%)	3228 (82.7%)	3406 (92.1%)	27.89 (6.18)	207 (7.2%)	2353 (57.9%)	809 (19.7%)
Became less diverse (n = 1294)	58.63 (10.09)	1035 (62.0%)	192 (15.0%)	1105 (87.9%)	1173 (91.7%)	27.61 (5.9)	70 (4.7%)	804 (63.2%)	228 (15.0%)
Became diverse (n = 1098)	59.91 (9.44)	885 (65.3%)	170 (16.2%)	957 (86.5%)	993 (94.2%)	27.15 (5.6)	57 (6.1%)	713 (66.7%)	195 (14.2%)
				Men					
Remained diverse (n = 1894)	60.29 (10.49)	1661 (74.5%)	459 (25.4%)	1717 (90.0%)	1716 (93.6%)	28.73 (4.89)	137 (9.7%)	1258 (63.3%)	515 (23.2%)
Remained less diverse (n = 3145)	58.08 (9.39)	2523 (65.5%)	634 (20.6%)	2752 (83.0%)	2810 (94.5%)	27.63 (4.59)	235 (9.0%)	2012 (63.1%)	1040 (33.1%)
Became less diverse (n = 832)	59.28 (10.37)	704 (71.7%)	208 (21.7%)	757 (90.8%)	756 (94.4%)	28.15 (4.53)	48 (6.4%)	529 (68.1%)	231 (22.5%)
Became diverse (n = 727)	59.53 (9.97)	624 (76.1%)	155 (21.7%)	673 (91.0%)	661 (95.7%)	28.11 (4.72)	54 (8.3%)	498 (62.1%)	220 (28.1%)

Descriptive statistics, including percentages (%) and means (SD), were calculated using CLSA survey inflation weights. All variables were reported at baseline, except for vegetable intake at follow-up 2. ^1^ The highest education level was post-secondary graduation [university degree]. ^2^ The highest income was ≥CAD 150,000.

**Table 3 nutrients-17-01005-t003:** Gender-specific odds ratios (95% CI) of non-daily vegetable intake associated with social isolation transitions in the Canadian Longitudinal Study on Aging (2011–2021).

	Model A		Model B		Model C		Model D	
	OR	CI95	OR	CI95	OR	CI95	OR	CI95
				Women				
Remained not isolated	Ref		Ref		Ref		Ref	
Remained isolated	2.10 ***	[1.51, 2.94]	1.85 ***	[1.32, 2.59]	1.85 ***	[1.31, 2.59]	1.85 ***	[1.32, 2.59]
Became isolated	1.32 *	[1.02, 1.69]	1.20	[0.93, 1.55]	1.19	[0.93, 1.54]	1.19	[0.93, 1.54]
Became not isolated	1.50 **	[1.13, 2.01]	1.36 *	[1.02, 1.82]	1.37 *	[1.03, 1.84]	1.38 *	[1.03, 1.84]
				Men				
Remained not isolated	Ref		Ref		Ref		Ref	
Remained isolated	1.32	[0.97, 1.80]	1.16	[0.85, 1.59]	1.14	[0.83, 1.56]	1.14	[0.83, 1.56]
Became isolated	1.25	[0.98, 1.61]	1.16	[0.90, 1.50]	1.16	[0.90, 1.49]	1.16	[0.90, 1.49]
Became not isolated	1.42 *	[1.07, 1.88]	1.32	[0.99, 1.75]	1.31	[0.99, 1.75]	1.31	[0.99, 1.74]

Gender-specific odds ratios (95% CIs) obtained by random coefficient logistic regression with an interaction term (social isolation and gender) on the sample with daily intake at baseline conditioning on confounders. Model A: Follow-up time, age, BMI, and chronic condition (n = 16,080). Model B: Model A + education, income, home ownership, and geographic location (n = 16,034). Model C: Model B + sleep deprivation (n = 15,999). Model D: Model C + provincial gross domestic product (GDP), % provincial food insecurity, vegetable consumer price index (CPI), and provincial public spending (n = 15,999). * *p* < 0.05; ** *p* < 0.01; *** *p* < 0.001.

**Table 4 nutrients-17-01005-t004:** Gender-specific odds ratios (95% CI) of non-daily vegetable intake associated with social participation transitions in the Canadian Longitudinal Study on Aging (2011–2021).

	Model A		Model B		Model C		Model D	
	OR	CI95	OR	CI95	OR	CI95	OR	CI95
				Women				
Remained diverse	Ref		Ref		Ref		Ref	
Remained less diverse	1.36 *	[1.20, 1.54]	1.29 *	[1.14, 1.47]	1.30 *	[1.15, 1.48]	1.30 *	[1.15, 1.47]
Became less diverse	1.12	[0.94, 1.33]	1.08	[0.91, 1.29]	1.08	[0.91, 1.29]	1.08	[0.91, 1.29]
Became diverse	1.12	[0.93, 1.34]	1.09	[0.91, 1.31]	1.10	[0.91, 1.32]	1.10	[0.91, 1.32]
				Men				
Remained diverse	Ref		Ref		Ref		Ref	
Remained less diverse	1.36 *	[1.20, 1.55]	1.29 *	[1.13, 1.46]	1.29 *	[1.13, 1.46]	1.29 *	[1.13, 1.46]
Became less diverse	1.04	[0.87, 1.25]	1.03	[0.85, 1.24]	1.03	[0.85, 1.24]	1.03	[0.85, 1.24]
Became diverse	1.18	[0.98, 1.43]	1.15	[0.95, 1.40]	1.16	[0.95, 1.40]	1.16	[0.95, 1.4]

Gender-specific odds ratios (95% CIs) obtained by random coefficient logistic regression with an interaction term (changes in diversity of social participation and gender) on the sample with daily intake at baseline conditioning on confounders. Model A: Follow-up time, age, BMI, and chronic condition (n = 16,080). Model B: Model A + education, income, home ownership, and geographic location (n = 16,034). Model C: Model B + sleep deprivation (n = 15,999). Model D: Model C + provincial gross domestic product (GDP, % provincial food insecurity, vegetable consumer price index (CPI), and public spending (n = 15,999). * *p* < 0.001.

**Table 5 nutrients-17-01005-t005:** Gender-specific odds ratios (95% CI) of non-daily fruit intake associated with social isolation transitions in the Canadian Longitudinal Study on Aging (2011–2021).

	Model A		Model B		Model C		Model D	
	OR	CI95	OR	CI95	OR	CI95	OR	CI95
				Women				
Remained not isolated	Ref		Ref		Ref		Ref	
Remained isolated	2.42 **	[1.72, 3.40]	2.19 **	[1.56, 3.08]	2.23 **	[1.58, 3.14]	2.23 **	[1.58, 3.14]
Became isolated	1.19	[0.90, 1.57]	1.14	[0.86, 1.50]	1.14	[0.86, 1.51]	1.14	[0.86, 1.51]
Became not isolated	1.88 **	[1.41, 2.51]	1.76 **	[1.32, 2.35]	1.77 **	[1.33, 2.37]	1.77 **	[1.33, 2.37]
				Men				
Remained not isolated	Ref		Ref		Ref		Ref	
Remained isolated	1.46 *	[1.06, 2.01]	1.35	[0.98, 1.87]	1.35	[0.98, 1.86]	1.35	[0.98, 1.86]
Became isolated	1.32 *	[1.03, 1.68]	1.28	[1.00, 1.63]	1.27	[1.00, 1.63]	1.27	[1.00, 1.63]
Became not isolated	1.21	[0.91, 1.62]	1.17	[0.88, 1.56]	1.17	[0.88, 1.57]	1.17	[0.88, 1.57]

Gender-specific odds ratios (95% CIs) obtained by random coefficient logistic regression with an interaction term (social isolation and gender) on the sample with daily intake at baseline conditioning on confounders. Model A: Follow-up time, age, BMI, and chronic condition (n = 18,494). Model B: Model A + education, income, home ownership, and geographic location (n = 18,448). Model C: Model B + sleep deprivation (n = 18,413). Model D: Model C + provincial gross domestic product (GDP), % provincial food insecurity, fruit consumer price index (CPI), and provincial public spending (n = 18,413). * *p* < 0.05; ** *p* < 0.001.

**Table 6 nutrients-17-01005-t006:** Gender-specific odds ratios (95% CI) of non-daily fruit intake associated with social participation transitions in the Canadian Longitudinal Study on Aging (2011–2021).

	Model A		Model B		Model C		Model D	
	OR	CI95	OR	CI95	OR	CI95	OR	CI95
				Women				
Remained diverse	Ref		Ref		Ref		Ref	
Remained less diverse	1.71 ***	[1.49, 1.96]	1.63 ***	[1.42, 1.87]	1.64 ***	[1.43, 1.88]	1.64 ***	[1.43, 1.88]
Became less diverse	1.37 **	[1.14, 1.65]	1.34 **	[1.12, 1.62]	1.35 **	[1.12, 1.62]	1.35 **	[1.12, 1.62]
Became diverse	1.24 *	[1.01, 1.51]	1.21	[0.99, 1.48]	1.22	[1.00, 1.49]	1.22	[1.00, 1.49]
				Men				
Remained diverse	Ref		Ref		Ref		Ref	
Remained less diverse	1.32 ***	[1.16, 1.51]	1.28 ***	[1.12, 1.46]	1.28 ***	[1.12, 1.46]	1.28 ***	[1.12, 1.46]
Became less diverse	1.07	[0.89, 1.30]	1.07	[0.88, 1.29]	1.07	[0.88, 1.29]	1.07	[0.88, 1.29]
Became diverse	1.10	[0.90, 1.34]	1.08	[0.88, 1.32]	1.07	[0.87, 1.30]	1.07	[0.87, 1.30]

Gender-specific odds ratios (95% CIs) obtained by random coefficient logistic regression with an interaction term (changes in diversity of social participation and gender) on the sample with daily intake at baseline conditioning on confounders. Model A: Follow-up time, age, BMI, and chronic condition (n = 18,494). Model B: Model A + education, income, home ownership, and geographic location (n = 18,448). Model C: Model B + sleep deprivation (n = 18,413). Model D: Model C + provincial gross domestic product (GDP, % provincial food insecurity, fruit consumer price index (CPI), and public spending (n = 18,413). * *p* < 0.05; ** *p* < 0.01; *** *p* < 0.001.

## Data Availability

Data are available through a formal data access request to the Canadian Longitudinal Study on Aging.

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
