# Peer review of "Gender, Adverse Changes in Social Engagement and Risk of Unhealthy Eating: A Prospective Cohort Study of the Canadian Longitudinal Study on Aging (2011–2021)"

_nutrients, 2025, doi:10.3390/nu17061005_

Round 1

Reviewer 1 Report

Comments and Suggestions for Authors

Reaserch concerns the important problem of aging, which can be accelerelated  or delayed by environmental factors. A particularly, unfavorable  factor is improper nutrition, which can be changed in various ways, such as improving informations, programmatic training, production of food for  the elderly,   with the necessary microelements and other. I suggest the more attention be paid to these issues in the discussion chapter.

Reviewer 2 Report

Comments and Suggestions for Authors

Interesting work, referring to current research topics. 

Introduction 
The study objective presented in the Introduction (lines 80-85) reiterates information already covered in the Introduction, where the authors discussed the impact of social participation on healthful eating habits and highlighted the research gap regarding the lack of studies examining changes in social participation about dietary habits, particularly concerning gender differences. To avoid redundancy, the study objective should focus on a more precise formulation of the hypothesis and its testing approach rather than restating the theoretical background.

Materials and methods 
Lines 100-104. I propose to describe more clearly, e.g., by separating information on the number of people eligible for the study from the number of people included in the analysis. 

Point 2.2. 
It is not clear why the category of "less healthy eating" includes only a reduction in the frequency of consumption to less than once a day. Were other aspects of the diet also analyzed, e.g., increased consumption of processed foods? 
The authors reported that fruit and vegetable consumption outliers were excluded based on the National Cancer Institute Dietary Screener Questionnaire (NHANES 2009–2010). I propose to add a justification for these thresholds (e.g., why >8 times a day for fruit, >5 times a day for vegetables, and >3 times a day for potatoes considered unrealistic?). 
It was described that the vegetable variable was coded as "daily consumption" based on any of the four types of vegetables (greens, potatoes, carrots, and other vegetables). Some doubts exist, e.g., if a participant only ate potatoes daily, was he still classified as a healthy eater? 

Point 2.3. 
The authors defined regular participation in social activities as participation at least once a month and less frequent participation as less than once a month. It is interesting why once a month was adopted as a criterion – is this not too broad a criterion, considering the differences in the nature of social activities? For example, regular participation in physical activity has a different meaning than participation in social gatherings.
 Point 2.6
 The description of statistical methods is generally well described. The reviewer's doubts concern:
 - lack of explanation for CLSA survey inflation weights – Although their use in descriptive analyses is mentioned, it would be worth the authors to clarify why they were not included in inferential analyses (it was only said that CLSA does not provide weights at the individual and geographical level). Does this affect the interpretation of the results? 
- selection of variables in sensitivity analyses – The authors listed additional variables but did not indicate why these variables (e.g., oral health, physical activity, smoking) were included as potential confounders. Is this based on previous studies? 

Results 
The gender results could be better highlighted in the text, e.g., by introducing explicit summaries for women and men.

Discussion 
The discussion is well described in the context of previous studies, indicating consistency with existing literature and new conclusions. The authors present key findings, including gender differences in the impact of social isolation and social participation on healthy eating. I propose to emphasize the practical use of the results more, e.g., in health policy or social interventions. 

Conclusions 
I propose to rephrase the last sentence, as it is difficult to understand.

Reviewer 3 Report

Comments and Suggestions for Authors

Dear Authors,

It was a pleasure to conduct the peer review of your manuscript, which, in my opinion, addresses an interesting topic and demonstrates a high methodological quality. I would like to commend you on the excellent work.

Below are some revisions and suggestions:

  • One aspect that would be interesting to further explore is the impact of social isolation during the COVID-19 pandemic, which may have amplified effects on fruit and vegetable intake. Although the observation period covers the years 2011-2021, including the specific dietary and social changes during the pandemic would be crucial to fully understand the impact of extraordinary events on eating behavior.

I suggest adding a section in the discussion that explores how social isolation during the pandemic may have influenced the results, proposing that the lockdown period exacerbated unhealthy eating behaviors due to the further reduction in social interactions and the increase in sedentary activities. These aspects could be discussed in light of other studies that have addressed this topic, such as those related to diet and eating habits during the pandemic. I recommend comparing this aspect with the following articles: doi: 10.1016/j.jand.2022.08.132, doi: 10.3390/medicina60101624, and doi: 10.1016/j.appet.2024.107727.

  • Although the results show a strong impact of social changes on fruit and vegetable intake in women, it would be interesting to explore further the reasons behind these gender differences. The discussion mentions that women may have wider social networks, but economic, psychological, and cultural factors might also play a role in the differing impact of social isolation on men and women. A theoretical exploration of how gender roles and social expectations influence eating behaviors would be useful to enrich the analysis.
  • The manuscript acknowledges the potential recall bias related to the collection of dietary and social activity data. It would be appropriate to suggest, as indicated for future studies, the implementation of methodologies to reduce this bias, such as the use of more objective monitoring technologies (e.g., apps for recording eating and social behaviors) or a more quantitative analysis of dietary data.

The article is well-written, with a solid methodological analysis and a good discussion of the results. The strengths lie in the innovation of the topic and the robustness of the methods used.

However, I suggest making the work even more comprehensive and relevant by specifying in the methods section of your study which reporting guideline was followed, specifically that of the EQUATOR NETWORK for the type of study design adopted. Additionally, the authors should provide the Editor with the corresponding checklist, as required by the reporting guidelines.

Align the bibliography according to the journal guidelines

Reviewer 4 Report

Comments and Suggestions for Authors

dear, thank you for your work. I read your paper with interest. part of your references are older than 10 years. can you change them with the newest (max 5 years)?

Round 2

Reviewer 3 Report

Comments and Suggestions for Authors

I commend the authors on their work. The manuscript is well-prepared and ready for publication.